DOI: 10.1038/s41467-018-03268-y　　OPEN

# Highly stretchable carbon aerogels

Fan Guo[1], Yanqiu Jiang[1], Zhen Xu [1], Youhua Xiao[1], Bo Fang[1], Yingjun Liu[1], Weiwei Gao[1], Pei Zhao [2], Hongtao Wang[2] & Chao Gao[1]

Carbon aerogels demonstrate wide applications for their ultralow density, rich porosity, and multifunctionalities. Their compressive elasticity has been achieved by different carbons. However, reversibly high stretchability of neat carbon aerogels is still a great challenge owing to their extremely dilute brittle interconnections and poorly ductile cells. Here we report highly stretchable neat carbon aerogels with a retractable 200% elongation through hierarchical synergistic assembly. The hierarchical buckled structures and synergistic reinforcement between graphene and carbon nanotubes enable a temperature-invariable, recoverable stretching elasticity with small energy dissipation (~0.1, 100% strain) and high fatigue resistance more than $10^6$ cycles. The ultralight carbon aerogels with both stretchability and compressibility were designed as strain sensors for logic identification of sophisticated shape conversions. Our methodology paves the way to highly stretchable carbon and neat inorganic materials with extensive applications in aerospace, smart robots, and wearable devices.

---

[1] MOE Key Laboratory of Macromolecular Synthesis and Functionalization, Department of Polymer Science and Engineering, Zhejiang University, 38 Zheda Road, Hangzhou 310027, China. [2] Institute of Applied Mechanics and Key Laboratory of Soft Machines and Smart Devices of Zhejiang Province, Zhejiang University, 38 Zheda Road, Hangzhou 310027, China. Fan Guo and Yanqiu Jiang contributed equally to this work. Correspondence and requests for materials should be addressed to Z.X. (email: zhenxu@zju.edu.cn) or to C.G. (email: chaogao@zju.edu.cn)

Carbon aerogels (CAs) have characteristics of ultralow density, rich porosity, high conductivity, and extreme environmental stabilities[1–5], which allow wide applications such as damping components[2, 3], environmental protections[6–8], energy storages[9–13], sensors[14, 15], catalysts[16, 17], and electromagnetic metamaterials[18, 19]. However, monolithic CAs have suffered from their poor mechanical strength because of the extremely dilute connections and fragile joints in their porous network[20]. In the past decade, the compressive brittleness has been well resolved, affording various super-compressible CAs[21–28]. Nonetheless, CAs still behave severely brittle under tensile deformation[29], limiting their uses for growing demands in stretchable electronics, wearable devices, and smart manufacturing.

In the pursuit of CAs with better mechanical robustness, two main strategies have been exploited to amend their tensile brittleness[30–39]. The prevailing approach is the introduction of elastic polymers[30–33]. High tensile elongation is easily achieved by virtue of elastomers, but this elasticity is derived from the intrinsic entropic elasticity of polymers, which is less stable in harsh chemical and physical surroundings. For example, the elasticity becomes brittle under low temperature for the frozen chains and becomes viscous at high temperature. This method just brings polymer elasticity to CAs but the inherent tensile brittleness of neat CAs is still unsolved. Further, blending with polymers weakens the favorable functionalities of CAs such as highly electrical conductivity and low density. Another approach is to enhance interconnections of CAs[34–39]. For instance, Wang et al.[34] fabricated all-carbon conductors from multiwalled carbon nanotubes (MWNTs), with joints welded by amorphous carbon. Despite tremendous efforts, the stretchable elongation was still limited under 25% and aerogels completely broke down upon higher strain. Therefore, achieving highly stretchable neat CAs is a big challenge unresolved yet.

Here we report a method of hierarchical synergistic assembly to achieve highly stretchable neat CAs. Our binary CAs (bCAs), consisting of graphene and MWNTs, are designed to have four orders of hierarchical structures ranging from nanometer to centimeter. The hierarchical structures and synergistic effect between graphene and MWNTs collectively enable the superb stretchability up to 200% elongation of neat bCAs with density down to 5.7 mg cm$^{-3}$. Moreover, minor plastic deformation (~1%), low energy dissipation (~0.1), excellent fatigue resistance ($10^6$ cycles), and environment stability (93–773 K) are achieved simultaneously for bCAs, which are better than those of silicon rubbers. The integration of three-dimensional (3D) ink-printing technique allows the design of lattice structure of bCAs and controls over mechanical deformation behaviors. The ultralight bCAs with both stretchability and compressibility are used as strain sensors for precise logic identification of complex shape conversions. Our assembly strategy opens the avenue to highly stretchable carbon and other neat inorganic materials for wide applications in aerospace, smart robots, and wearable devices.

## Results

### Fabrication and characterization of bCAs.
Our bCAs were fabricated by ink-printing homogeneous aqueous mixtures of graphene oxide (GO) and purified MWNTs, followed by freeze-drying and reduction under confined state (Fig. 1a). Different from previous reports[23, 40, 41], we added trace calcium ions (15 mM) as gelators to enable readily direct writing monolithic lattices under ambient surroundings (Supplementary Figs 1 and 2). Homogeneous GO-MWNT gel inks were extruded out through a movable nozzle (~250 μm of diameter) to additively deposit into program-controlled 3D structures. After freeze-drying, the resulting porous GO-MWNT aerogels were reduced by chemical or thermal treatments under pre-buckled states at a given compression ratio ($\alpha$, up to 70%) to get bCAs with ultralow density down to 5.7 mg cm$^{-3}$ (Fig. 1b).

The bCAs were designed to have four orders of structural hierarchy spanning from centimeter to molecular scale (Fig. 1c–f). The monolithic CAs encompassed macroscopic trusses (the first

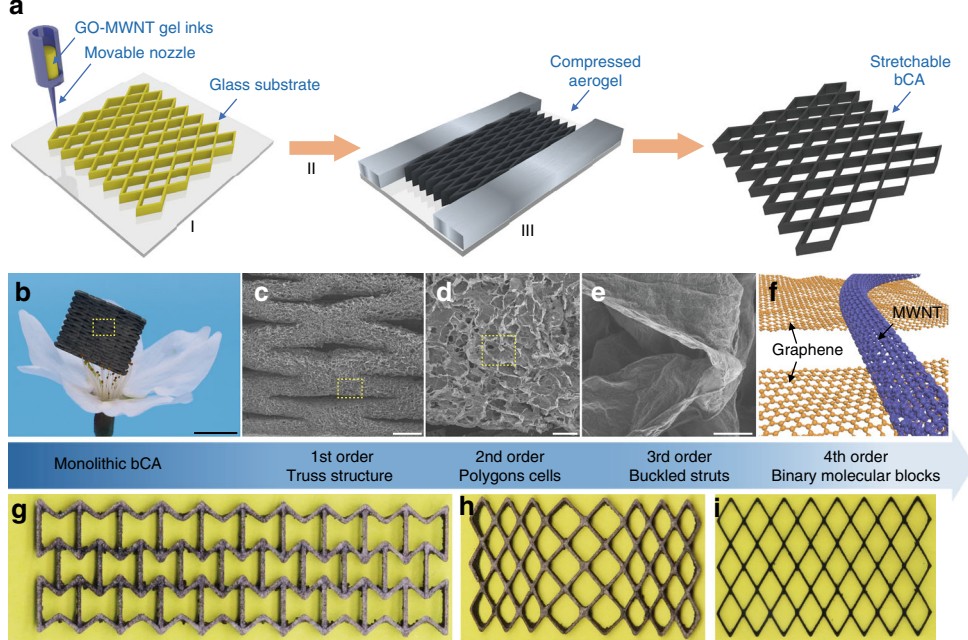

**Fig. 1** Design and hierarchical architecture of synergistic bCAs. **a** Schematic illustration of the hierarchical synergistic assembly for fabrication of stretchable bCAs through 3D printing (I) followed by freeze-drying (II) and pre-buckled reduction (III). **b** A digital photograph of ultralight bCAs with density of 5.7 mg cm$^{-3}$ floating on a flower. **c–f** SEM images of quaternary structure of bCAs across multi-size scales. **g–i** Printed lattices by design with negative (−0.3, **g**), positive (+0.5, **h**), and nearly zero Poisson ratio (**i**). Scale bars, 5 mm (**b**), 500 μm (**c**), 100 μm (**d**), and 5 μm (**e**)

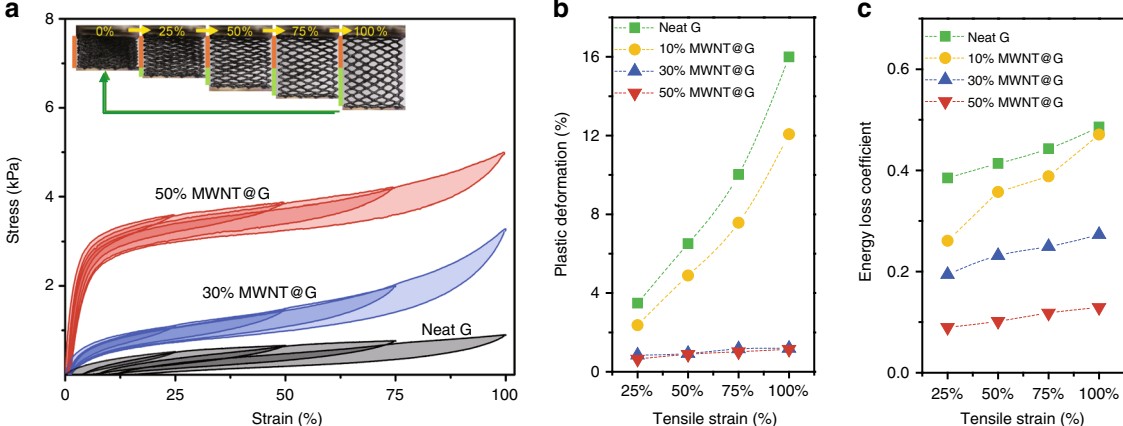

**Fig. 2** Uniaxial monotonic tensile tests. **a** Stress-strain curves of bCAs with increasing dose of MWNTs at 25, 50, 75, and 100% tensile strains. **b**, **c** Plastic deformation and energy loss coefficient corresponding to different stains in **a**

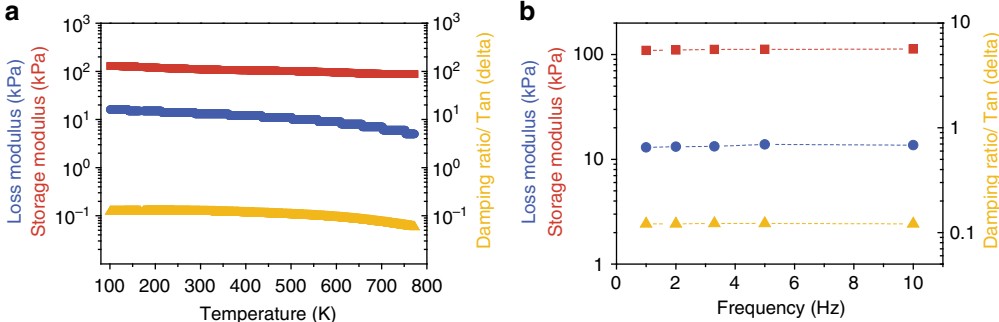

**Fig. 3** Mechanical stability of bCAs. **a**, **b** Storage modulus (red), loss modulus (blue), and damping ratio (yellow) as function of temperature (**a**) and frequency (**b**) of bCAs with 30 wt% MWNTs

order), voronoi polygon cells enclosed by MWNT-interconnected graphene laminates (the second order in dozens of microns), folded cell walls that might allow reversible stretching (the third order in microns), and synergistic binary molecular building blocks (the fourth order). Such a hierarchical integration facilitates high stretchability of bCAs. From an energetic standpoint, the restored conjugated domains of reduced graphene enhanced the van der Waals interaction with MWNTs, and the stored work in compression and chemical energy in reduction provided the elastic strain energy to conform to large tensile deformations (Supplementary Figs 3 and 4). As shown in Supplementary Fig. 5, reduced strip-shaped aerogels have higher breaking strength than unreduced counterparts. The programmed direct writing allowed the design of bending-dominated macro-lattices to control their tensile behaviors and Poisson ratios, e.g., the re-entrant auxetic honeycomb with a negative Poisson ratio in Fig. 1g, positive in Fig. 1h, and almost zero in Fig. 1i (Supplementary Fig. 6).

**Monotonic tensile tests**. The highly reversible stretchability of bCAs was demonstrated in monotonic and cyclical tensile tests on monolithic samples (1.0 cm the gauge length). The monotonic tensile tests with progressively increasing strains (Fig. 2a) revealed that the plastic deformation decreased dramatically as MWNTs dose increased, from 16–60% for neat graphene aerogels (Supplementary Fig. 7) to 1% for the samples containing MWNTs higher than 30 wt% (Fig. 2b). The Young's modulus and tensile strength of bCAs were enhanced by about 20 and 5 times, respectively, compared with neat graphene aerogels. For an elastic

material, energy dissipation, stemming from the interior fiction and localized cracks, is evaluated by the energy loss coefficient ($\gamma$, the area ratio of hysteresis loops to tensile curves)[42]. The bCAs with 50 wt% MWNTs attained a particularly low $\gamma$ down to 0.1, only a quarter of that of neat graphene counterparts (Fig. 2c), outperforming both carbon nanotube (CNT) foams ($\gamma$, 0.6 at 57% compressive strain)[3] and commercial silicon rubbers ($\gamma$, ~0.3 at 100% tensile strain, Supplementary Fig. 8b). These results indicate that the integration of graphene and MWNTs considerably improves the stretching elasticity of bCAs.

Notably, the stretching elasticity of bCAs with 30 wt% MWNTs kept invariable in a wide spectrum of frequency (1–10 Hz) and wide range of temperature (at least 93–773 K) in Fig. 3a, b. This energetic elasticity is more stable than entropy elastic polymer rubbers, because of strong atomic bonding in graphene and MWNTs. For instance, the elasticity of polymer rubbers is dependent on both frequency and temperature (e.g., the commercial silicon rubber becomes brittle below 150 K)[42, 43].

**Fatigue resistance tests**. The fatigue resistance was further evaluated by long cyclical tensile tests (Fig. 4a and Supplementary Table 1). The bCAs with MWNTs higher than 10 wt% exhibited a very small plastic deformation (~1%) at the first 100% strain cycle due to the release of internal stress (Supplementary Movie 1), and then the following cycles showed identical curves in tested 100 cycles (Fig. 4b). For comparison, the control sample of neat graphene aerogel and the commercial silicon rubber showed large plastic deformations during the multi-cycles about ~40–65% and 6–10%, respectively. As MWNTs dose increased, the $\gamma$ of bCAs

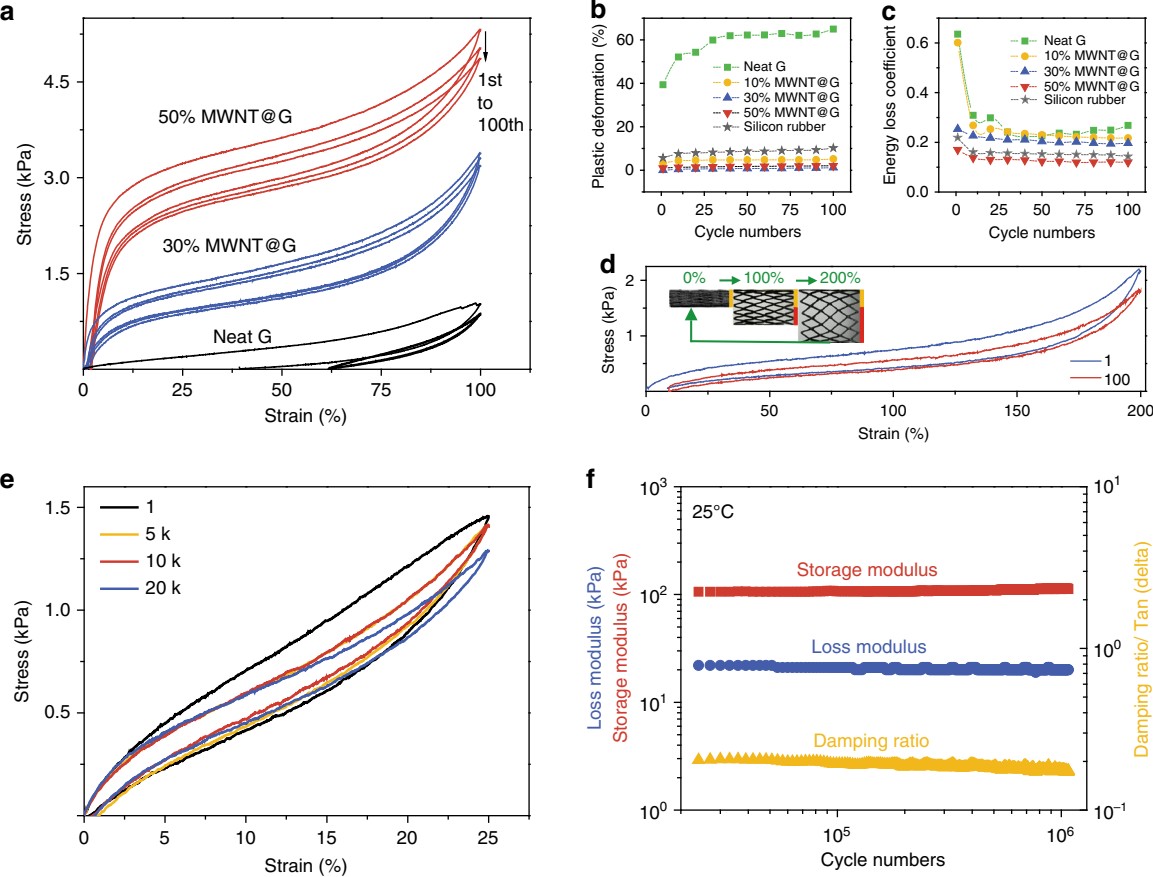

**Fig. 4** Fatigue resistance tests. **a** Stress-strain curves of bCAs with increasing dose of MWNTs at 100% tensile strain for 100 cycles (density: 9.7 mg cm$^{-3}$). **b**, **c** Plastic deformation and energy loss coefficient ($\gamma$) during the first 100 tensile cycles in **a**. **d** Cyclical tensile-release curves for a designed bCA with stretchable ratio of 200%. **e** Cyclical stretching curves for 20 000 times at 25% stain at the rate of 0.01 s$^{-1}$. **f** Fatigue resistance of bCA during 10$^6$ stretching cycles at 1% strain by DMA at 100 Hz

monotonously decreased from 0.3–0.6 of neat graphene samples to ~0.15 of counterparts with 50 wt% MWNTs (Fig. 4c). These results imply almost no structure failure during the repeating stretching of bCAs. The bCAs even remained good fatigue resistance at a remarkable 200% tensile strain during 100 cycles (Fig. 4d). We further took tensile tests and dynamic thermomechanical analysis to assess their fatigue resistance in extremely long cycles. The bCAs with 30% MWNTs displayed nearly identical curves over 20 000 cyclical stretching-release tests at a large 25% strain (Fig. 4e). Under high frequency of 100 Hz (100 deformation cycles in 1 s), the corresponding storage and loss moduli remained highly stable over 10$^6$ cycles at ~10% strain, and the damping ratio kept stable at a small value of 0.2 (Fig. 4f), much lower than those of CNT cottons (0.3)[42].

**Synergistic hierarchical structures**. The structural tracking reveals that the highly stretchable elasticity of bCAs originates from the combination effect of the multi-order hierarchical structures and synergistic reinforcement between MWNTs and graphene. In the first order, the millimetric hinged struts rotate by joints under tension. During a 100% repeating stretching-recovery deformation, the angle between two adjacent strut arms opened to 60° and recovered to 10° after released. This rotation of struts generated a local 450% elongation between joints, from 140 to 770 μm (Fig. 5a), and finally induced a 100% monolithic strain of bCA. The rhombic millimeter lattice effectively amplifies local deformations of joints at a tunable ratio.

Geometrical analysis demonstrates that the stretchable ratio ($\beta$) of bCAs can be designed by controlling the anisotropy of struts as $\beta \sim \frac{l}{2d} \cdot \sin\theta$, where $l$ and $d$ are the length and diameter of arms, respectively, and $\theta$ is the sharp degree of angles that open as stretched (Supplementary Fig. 9). Therefore, we enlarged the anisotropy of struts and a 200% reversible strain of bCAs was facilely achieved. By contrast, bulk binary aerogels prepared by the same process as in the case of bCAs only presented a limited stretchable elasticity (<20% strain) because of the absence of hierarchy in multi-orders (Supplementary Fig. 10).

We further examined the strain-concentrated joints in tensile cycles by in situ scanning electron microscope (SEM) inspections. The second-order synergistic binary polygon cells together with third-order buckled cell walls enabled the reversible tensile deformation. When a tensile force was loaded, the 3D polygons expanded along the tensile direction (Fig. 5b), by virtue of reversible deformations of cell walls in multimodes, including bending, torsion, extension, and displacement (Supplementary Fig. 11)[44]. High-resolution examinations demonstrated that folded walls with undulated conformations were gradually unfolded and extended to a flat shape under tensile and recovered to the original wrinkled conformation as released (Fig. 5c). In the in situ transmission electron microscope (TEM), we observed a tension-to-torsion mechanism for a single synergistic flake of MWNT-covered graphene. Reversible torsion and extension in 3D space were clearly visualized in the maximum tensile distance of 700 nm (Fig. 5d, Supplementary Fig. 12, and Movie 2). The intertwined MWNTs network as reinforced rebars uncoiled and

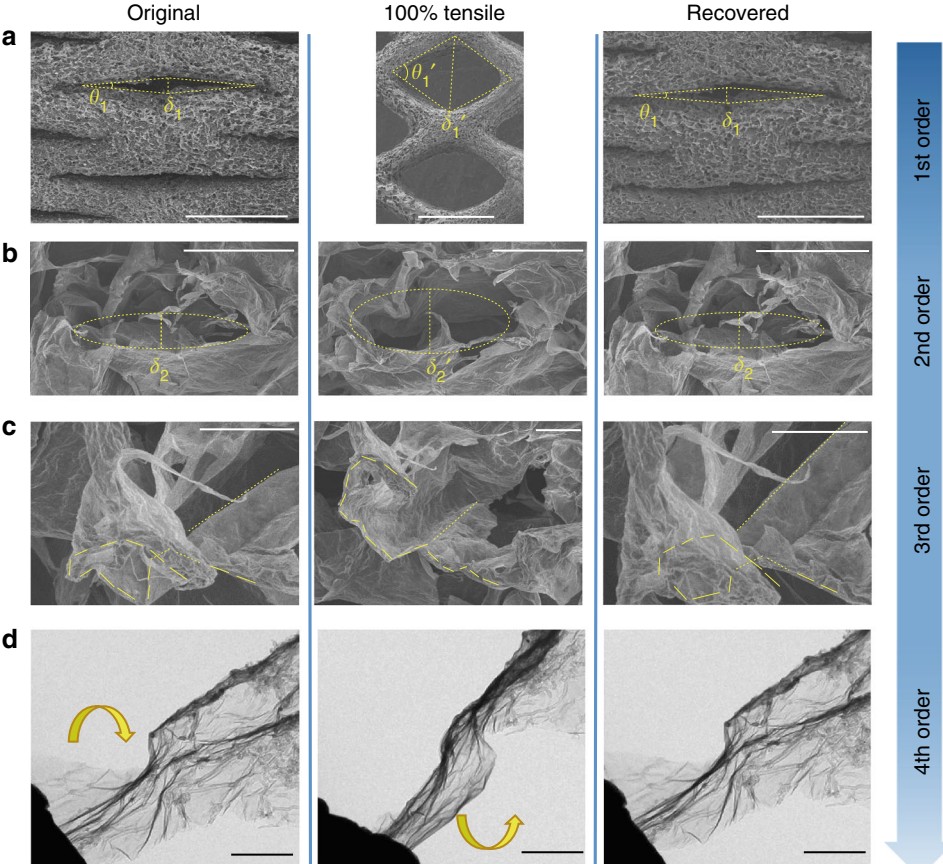

**Fig. 5** The bCA exhibits recoverable tensile deformation in four orders of hierarchy. **a–c** In situ SEM tracking on structural evolution of each orders structure in one cycle. **a** The length (140 μm, labeled by $\delta_1$) was stretched to 770 μm ($\delta_1'$) and recovered to 155 μm. **b** The length of a cell (9.7μm, $\delta_2$) was stretched to 17.6 μm ($\delta_2'$) and recovered to 10.1 μm. **d** In situ TEM observation of a binary laminate in one stretching-retraction cycle. Scale bars, 1 mm (**a**), 30 μm (**b**), 10 μm (**c**), and 100 nm (**d**)

twisted with graphene sheets as an integrated units and no sign of detachment was found. This observation implies that the tight attachment between graphene and MWNTs favors the efficient load transfer by a synergistic effect[45–47]. This synergistic effect may stem from the enhanced van der Waals force between graphene and MWNTs after chemical reduction, and almost no chemical bonds formed between carbon moieties at this mild reduction stage[48, 49]. The multi-order deformation process is schematically illustrated in Supplementary Fig. 13.

**Synergistic effect between graphene and MWNTs.** To reveal the root of synergistic effect, we used a strain-induced buckling metrology to assess the enhancement effect of MWNTs[50]. Thin binary films with thickness <120 nm were transferred onto the surface of polydimethylsiloxane slabs (Supplementary Fig. 14). The buckling instability during stretching generated regular wave-like wrinkles parallel to the tensile direction. The wavelength (λ) and thickness (h) of wrinkles of top films can couple with their elastic modulus as

$$\frac{E_f}{(1-V_f^2)} = \frac{3E_s}{(1-V_s^2)} \left(\frac{\lambda}{2\pi h}\right)^3 \qquad (1)$$

where $E_f$ and $E_s$ are the elastic moduli of binary film and substrate, respectively. $V_f$ and $V_s$ are Poisson's ratios thereof. The relative modulus of MWNT-graphene layers can be calculated as a function of $E_f = A \cdot \left(\frac{\lambda}{h}\right)^3$, where $A$ is a constant. From the statistics of widening spacing between wrinkles from optical

microscopy and SEM observations (Supplementary Fig. 15), the relative modulus was greatly enhanced by 100%, 250% and 150% as the dose of MWNTs increased to 10, 30, and 50 wt%, respectively (Fig. 6a). The enhancement in modulus explained the promoted tensile strength (~5 times) and super-elasticity of bCAs.

The cooperative effect of MWNTs also reflected in the improved fatigue resistance of bCAs. We measured the fatigue resistance of isolated synergistic flakes by the cyclical nanoindentation technique[51]. The constituent flakes were floated over circular copper holes (average diameter of 1 μm; Fig. 6b, c, insets) and cyclic compressions were taken at a maximum force of 64 nN. The irreversible displacement (40.9%) of 30 wt% MWNT binary flakes after 20 cycles was much smaller than that (64.5%) of neat graphene laminates (Fig. 6b, c). The strong binding between MWNTs and stacking graphene can relieve their plastic sliding and toughen graphene laminates. Analysis on facture fronts (Supplementary Figs 16, 17) revealed that MWNTs bond graphene sheets together to prevent the crack propagation by an extrinsic toughening mechanism and lock the joints to resist brittle breakage[52].

**Stretchable bCA-based strain sensors.** The self-standing bCAs with high electrical conductivity (~1000 S m$^{-1}$ after 3000 K thermal annealing) can be reversibly stretched (positive strain) and compressed (negative strain). For a bCA with a designed strain from +100% to −20%, the resistance gently increased (+10% at +100% strain) under stretching and steeply decreased

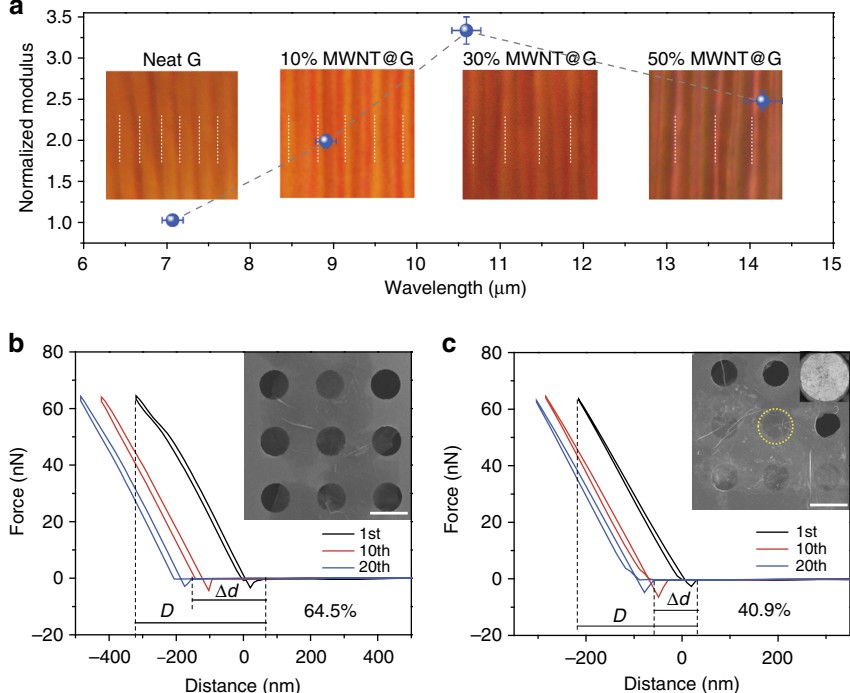

**Fig. 6** The enhancement effects of synergistic graphene and MWNTs. **a** Normalized modulus of graphene laminates with increasing MWNTs doses. Error bars represent the s.d. of wavelengths of different graphene laminates and corresponding normalized moduli for at least five measurements. The insets correspond to the optical microscopy images of buckled wave-like wrinkles with increasing wavelength. **b**, **c** Force-displacement curves of pure graphene flake (**b**) and synergistic flake (**c**) during 20 cycles. The insets are SEM and TEM images of isolated flakes of carbon aerogels on copper grids. Scale bars, 1 μm (**b**, **c**)

(−80% at −14% strain) under compressing (Fig. 7a, b), due to the corresponding stretching-collapsing structural evolution of interconnected polygon cells. The resistance-change-pattern of bCAs kept nearly identical in the cyclic deformation for at least 100 times (Fig. 7c). Considering the ternary response states of resistance, self-standing bCAs can act as a new generation of strain sensor to express three logic digits. Specifically, 1, −1 (labeled by T), and 0 were identified by the increasing, decreasing, and holding of resistance in sequence. As a model, we attached three bCA sensors onto joints on one side of a snake-like robot as a sensor array to monitor its movement and configurations. Three states of the labeled joints were read from positive (1), zero (0), and negative (T) change ratios of resistance (Supplementary Fig. 18). For instance, four configurations, including straight, crescent, S, and inverse S shapes, were read from changing ratios of resistance of located bCA sensors as (0 0 0), (1 1 1), (1 1 T), and (T 1 1) (Fig. 7d). The bCA strain sensors with the capability for logic identification become an important supplement for commercial sensors that only feel one-way deformation.

## Discussion

The past decades have witnessed the achieving of highly compressible CAs, whereas highly stretchable neat CAs are deemed as impossible[20, 53]. Our hierarchical synergistic assembly strategy presents a versatile avenue to achieve the challenged stretchability of neat carbon porous materials. This method shows two prime advantages: first, this assembly strategy precisely controls over both hierarchical structures ranging from micro- to macro-scale and Poisson ratios of monolithic aerogels; second, ultralight CAs are directly printed by the commonly available 3D printing machines without any modifications at ambient surroundings, which is quite simple and efficient. Previously, the 3D printing

technology has been applied to get metallic and ceramic micro-lattices, whereas the stretchable elongation was limited under 25% and none of them showed elastic resilience for the lack of ductile building blocks[23]. As a result, the multi-scale conformation control of our assembly strategy collectively alleviates the fragility of brittle materials and implies a general possibility to accurately fabricate CAs and other inorganic materials with highly cyclic stretchability.

The additive-free and ultralight bCAs provide a host of merits. Different from polymer-infiltrated materials, our aerogels, integrated by graphene and MWNTs, kept the pure composition of $sp^2$-hybridized carbon, thus maintained highly electrical conductivity (up to 1000 S m$^{-1}$) in spite of ultralow density (~10 mg cm$^{-3}$), small energy dissipation, and outstanding mechanical stability in a wide temperature range of 93–773 K. The bCAs showed preponderant elongation (~200%) and fatigue resistance (10$^6$) compared with existing aerogels without elastic additives, which breaks down at high elongation (>25%). This is ascribed to two main factors: the topology mixing of graphene (two-dimensional) and MWNT (one-dimensional) that effectively reinforces the mechanical strength of graphene laminates, and the multi-scale hierarchical structures that cooperatively deform upon large tensile strains. With all these unique properties, the bCAs are highly promising in slap-up applications where multifold functions are needed. We designed a preliminary model of strain sensor to accurately identify sophisticated and continuous sharp changes of smart robots, which is quite difficult for traditional strain sensors.

In summary, we demonstrated a precisely controlled hierarchical synergistic assembly strategy to fabricate highly stretchable neat CAs. Our bCAs kept structural stability under both large dynamic tension and compression deformations, and simultaneously exhibited ultralow density, highly electrical

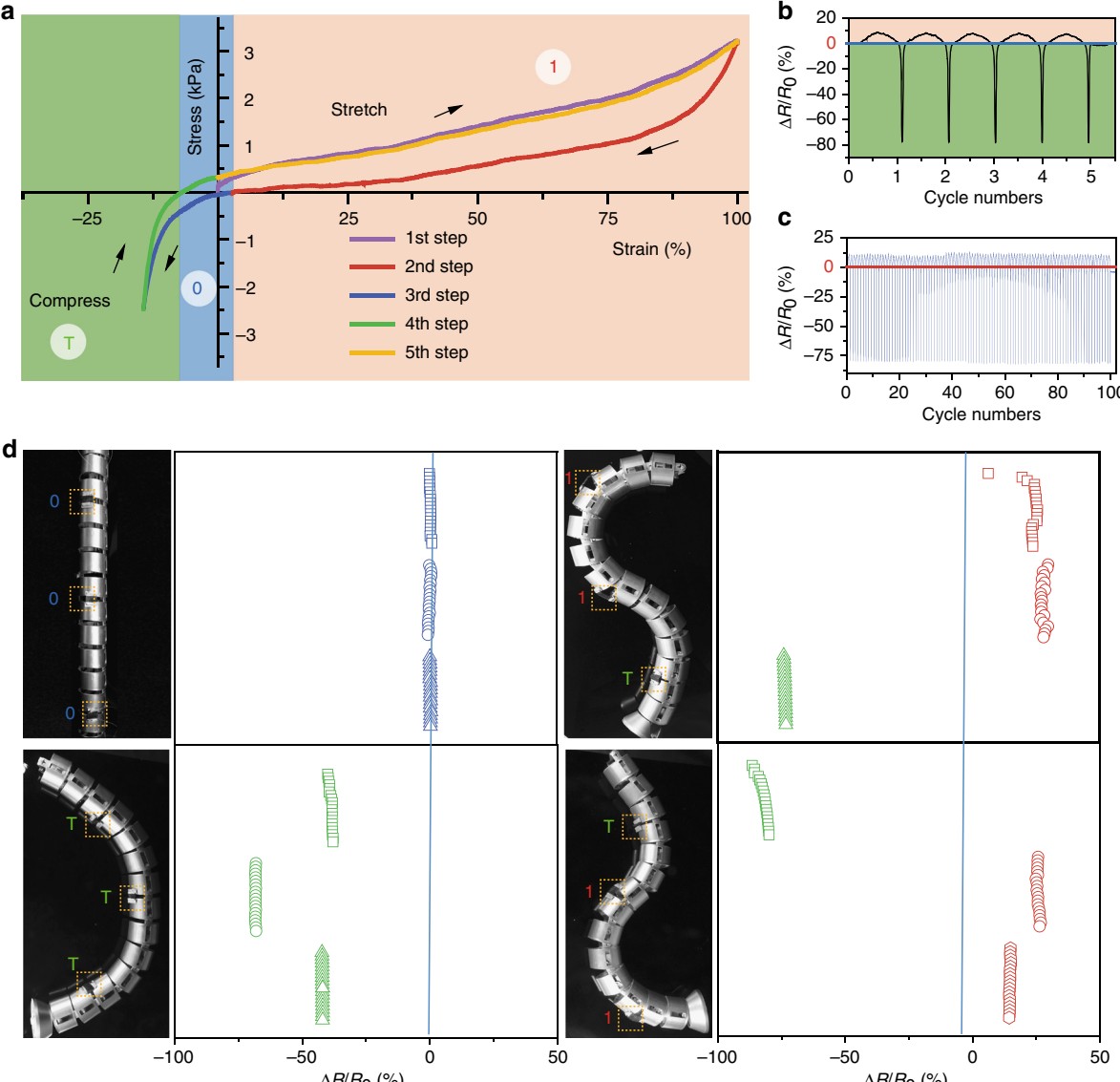

**Fig. 7** The bCAs as logic identification strain sensors. **a** Stress-strain curves of bCAs with 30 wt% MWNTs in a full strain range from +100% to −14%. **b, c** The evolution of resistance in the first 5 cycles (**b**) and 100 cycles (**c**) of bCAs. **d** A three-localized bCA strain sensor array works as the logic sensor to identify the movement and configuration of a snake-like robot

conductivity, and superb fatigue resistance. All these combined properties make our bCA adapted to high-end, various functions demanded applications, such as new strain sensors for logic identification, stretchable components to store and generate energy, and lightweight mechanical devices working in extreme temperature conditions. We believe that our assembly strategy can be possibly extended to fabricate other highly stretchable aerogels of inorganic constituents, opening a new era of rubber-like inorganic materials.

## Methods

**Fabrication of bCA monoliths**. Aqueous GO solution (20 mg ml⁻¹, lateral width of 10–20 μm) was purchased from Hangzhou Gaoxi Technology Co. Ltd (www.gaoxitech.com). The typical fabrication process of GO-MWNT (30 wt%) binary aerogel was described as follows: 0.5 g of 2.22 wt% CaCl₂ aqueous solution (11 mg CaCl₂) was gradually added to GO-MWNT mixed solution (GO ~ 7 mg ml⁻¹ and MWNTs ~ 3 mg ml⁻¹) with constant stirring, then a conditioning mixer (AR-100, THINKY) was used to further mix the ink at 3000 r.p.m. for 15 min. The prepared inks were used to fabricate 3D structure using a robotic deposition device (TH-206H, TIANHAO TECHNIC, China). The inner diameter of the printing nozzles

(d) is 250 μm. The 3D GO structures were printed onto glass substrate in air, with constant extruding pressure of 0.2 bar. The initial nozzle height from the substrate was set at around 0.8d to help the adhesion onto the substrate, and the moving speed of the nozzle ranges from 4 to 6 mm s⁻¹. The as-printed 3D GO aerogels were then frozen in liquid nitrogen and subsequently freeze-dried for 24 h. The as-prepared binary aerogels were compressed to a certain ratio and then reduced by reducing gas (HI or N₂H₄) or heat annealing at 3000 K for 30 min with a slow heating rate of 3 K min⁻¹ under inert gas.

**Characterizations**. Mechanical testing was taken on REGER-6000. The bCAs were stretched to specific elongation and then recovered to zero stress (no loading) or stretched to 25% tensile strain and back to zero strain (initial position). SEM inspections were taken on Hitachi S4800 field emission system. Raman spectra were taken on a Renishaw in Via-Reflex Raman microscopy at an excitation wavelength of 532 nm. X-ray photoelectron spectroscopy was measured using a PHI 5000 C ESCA system operated at 14.0 kV. All binding energies were referenced to the C1s neutral carbon peak at 284.8 eV. The compressive stress-strain tests were performed on a Micro-computer Control Electronic Universal Testing Machine (RGWT-4000–20, REGER). TEM images of synergistic flakes were taken on a JEM-2100 HR-TEM. Nanoindentation curves of flakes were taken in the tapping mode by carrying out on an Agilent 5500A atomic force microscope, with samples prepared by dropping on copper grids. X-ray diffraction data were collected on a X'

Pert Pro (PANalytical) diffractometer using monochromatic Cu 17 K$_{\alpha 1}$ radiation ($\lambda = 1.5406$ Å) at 40 kV.

To prepare sample for nanoindentation, reduced CAs with MWNTs (0–30 wt %) were evenly dispersed in dimethyl formamide (DMF) solution by ultrasonic cell disruption. Copper slabs with array of circular through-holes (diameters 1 μm) was used as substrate. Synergistic graphene/MWNT flakes in DMF dispersions were then dropped onto the copper substrate. With flakes entirely covering the holes, the tip of cantilevers was placed at the center of the hole. Two samples were both inflicted controlled force (about 64 nN) in order to compare the ratio ($\kappa$) between unrecovered distance ($\Delta d$) and entire deformation distance ($D$) $k = \Delta d/D$.

**Data availability**. The data that support the findings of this study are available on request from the corresponding authors (C.G. or Z.X.).

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

## Acknowledgements

This work is supported by the National Natural Science Foundation of China (Nos. 21325417, 51533008, 51703194, and 51603183), National Key R&D Program of China

(No. 2016YFA0200200), Hundred Talents Program of Zhejiang University (188020*194231701/113), National Postdoctoral Program for Innovative Talents (No. BX201700209), China Postdoctoral Science Foundation (2017M620241), and Fundamental Research Funds for the Central Universities (2017QNA4036).

## Author contributions

Z.X. and C.G. conceived the research. F.G, Y.J., and Z.X. designed experiments, analyzed the data, and wrote the manuscript. F.G., P.Z. and H.W. did the in situ TEM analysis of graphene laminates. F.G., B.F., Y.L., and W.G. did the electrical test. Y.X. purified MWNTs. Z.X. and C.G. oversaw all research phases. All authors discussed and commented on the manuscript.

## Additional information

**Competing interests:** The authors declare no competing interests.

