## [Peer Review File · Nature Communications]

Reviewers' comments:

Reviewer #1 (Remarks to the Author):

1. The authors employed a 3D printing method to make carbon (graphene and CNTs) aerogels. The technique has been reported in the literature. Two references (not authored by this reviewer) are given below, which the authors did not cite in their manuscript.

"3D Printing Hierarchical Silver Nanowire Aerogel with Highly Compressive Resilience and Tensile Elongation through Tunable Poisson's Ratio", *Small*, 2017

"3D Printing of Graphene Aerogels", *Small*, 2016.

2. Repetition of results. Figure 2(a) is identical to Fig.4(a). One should be removed.

3. Inconsistency of data: The data in Figure 4(e) do not seem to agree with any of those shown in Fig.4(a). Please explain or provide more details.

4. Significance of the findings: the results show that the aerogel can be stretched to 200% but exhibit strong nonlinear deformation. What is the significance or usefulness of this finding?

Reviewer #2 (Remarks to the Author):

This work designed and obtained a highly stretchable graphene foam material, by hierarchical structural design with the addition of CNT. Indeed, previously, there are many reports about graphene foams which demonstrated great elastic compressibility. While the more important and challenging stretching elasticity has been long sought, but it has yet to be achieved. With this, this report fills the important gap for graphene bulk material. Based on these, I would recommend this work to be accepted, with issues below to be addressed:

1. Importantly, the authors have put lot of effort to explain the reasons and mechanism for the highly temperature-invariable, recoverable stretching elasticity in Fig 5 etc. But I feel one important question needs to answer is the connection/bonding nature at the molecular level between graphene/CNT, graphene/graphene sheets, etc? I suspect that there are some types of chemical bonding between graphene/graphene, graphene/CNT, generated during the hydrothermal and annealing process. This has been both theoretically proposed and experimentally demonstrated, see for examples: *PHYSICAL REVIEW B* 74, 214104, 2006 and *Nature Comm*, 2015, 6, 6141. This should be the main reason for the structural stability under both stretching and compression.

2. Also, what is the role of Ca⁺⁺? What are the comparison without Ca⁺⁺?

3. Some statements need to be rechecked and more accurate, one of the examples is "However, monolithic CAs are prone to lose structural integrity under deformations in either tensile or compression mode because of the extremely dilute connections and fragile joints in their porous network", this is obviously not accurate since there are excellent results about the structural stability of graphene foam under even high compression (high elasticity), see *Nature Comm*, 2015, 6, 6141, *Adv. Mater.*, 2016, 28, 3504, and related papers. I feel the refs about this part is over missed.

Reviewer #3 (Remarks to the Author):

Improving the stretchability of carbon aerogels is one of the areas that are very important for several applications, namely in aerospace, smart robots, and wearable devices, among many others. Their use is however limited by the brittleness under tensile deformation. In addition, obtaining reversible extensive stretchability of such aerogels is also an unsolved problem.

This is extremely well written, interesting and thoughtful paper. Experiments are well conducted and the results are well documented. The conclusions are sound. The authors of the paper have clearly done a lot of work, and the results are entirely interesting.

The paper describes new way of making highly stretchable carbon aerogels using hierarchical synergistic assembly. These novel aerogels offer exceptional properties, very well described in the paper. The incorporation of 3D printing permits the design of various lattice structures, thus enabling different mechanical properties to be obtained. The described strategy of making these aerogels also enable detailed control of the hierarchical synergistic assembly.

These results are important and the more so as they seem to involve different way of approaching the fabrication of carbon aerogels, unlike all other approaches (techniques) used and described before in the literature. The study has clearly very good practical significance. I have no hesitation in recommending this paper for publication.

Responses to Reviewers

For Reviewer #1:

Comment 1: The authors employed a 3D printing method to make carbon (graphene and CNTs) aerogels. The technique has been reported in the literature. Two references (not authored by this reviewer) are given below, which the authors did not cite in their manuscript. "3D Printing Hierarchical Silver Nanowire Aerogel with Highly Compressive Resilience and Tensile Elongation through Tunable Poisson's Ratio", Small, 2017. "3D Printing of Graphene Aerogels", Small, 2016.

Response: Thanks for your kind reminder. We added these two references in the revised manuscript (ref. 40, 41).

Comment 2: Repetition of results. Figure 2(a) is identical to Fig.4 (a). One should be removed.

Response: Thanks for the comment. These two figures revealed different aspects of tensile property, both providing valuable information of the elastic behavior of our bCAs. In figure 2a, the bCAs are monotonically stretched to different strains (25% to 100%) to investigate the deformation characteristics at a single round of stretching. In figure 4a, the bCAs are stretched to 100% tensile strain and released for 100 cycles to evaluate their fatigue resistance performance.

Comment 3: Inconsistency of data: The data in Figure 4(e) do not seem to agree with any of those shown in Fig.4 (a). Please explain or provide more details.

Response: Thanks for the comment. The distinction between the two figures is mainly attributed to the first cycle, which comes from different testing samples and methods. (1) The internal stresses of any two bCAs (with same dose of

MWNTs) are slightly different, which will impact the shape of the first curve. The internal stress is released after the first cycle and the shapes of the curves tend to be steady. If we compare the curves since the second cycle, they will be quite similar. (2) In figure 4a, the bCAs are stretched to 100% elongation and then recovered to zero stress (no loading), which makes a complete cycle. In figure 4e, the bCAs are stretched to 25% tensile strain and back to zero strain (initial position). As a result, the curves at the origin seem different. We have added the detailed testing methods in characterizations (page 14).

Comment 4: Significance of the findings: the results show that the aerogel can be stretched to 200% but exhibit strong nonlinear deformation. What is the significance or usefulness of this finding?

Response: Thank you for the comment. The significance of our work is the achievement of highly stretchable and anti-fatigue aerogels based on brittle carbon materials. It is true the deformation of our bCAs is nonlinear, which is similar to that of polymer elastomers. Please note that, the deformation of previously reported compressible carbon aerogels is also nonlinear. As you insightfully pointed out, the deformation models could be controlled by the design of microlattices, which is being investigated in our lab now.

For Reviewer #2:

General comment: This work designed and obtained a highly stretchable graphene foam material, by hierarchical structural design with the addition of CNT. Indeed, previously, there are many reports about graphene foams which demonstrated great elastic compressibility. While the more important and challenging stretching elasticity has been long sought, but it has yet to be achieved. With this, this report fills the important gap for graphene bulk material. Based on these, I would recommend this work to be accepted, with issues below to be addressed:

Response: Thanks for your positive comment.

Comment 1: Importantly, the authors have put lot of effort to explain the reasons and mechanism for the highly temperature-invariable, recoverable stretching elasticity in Fig 5 etc. But I feel one important question needs to answer is the connection/bonding nature at the molecular level between graphene/CNT, graphene/graphene sheets, etc? I suspect that there are some types of chemical bonding between graphene/graphene, graphene/CNT, generated during the hydrothermal and annealing process. This has been both theoretically proposed and experimentally demonstrated, see for examples: PHYSICAL REVIEW B 74, 214104, 2006 and Nature Comm, 2015, 6, 6141. This should be the main reason for the structural stability under both stretching and compression.

Response: Thanks for your advice. The chemically reduced carbon aerogels (no hydrothermal treatment and high temperature annealing) also showed highly stretchable performance as demonstrated in our manuscript, in which we think van der Waals forces dominated the interaction between carbon moieties, rather than chemical bonding. We will try to introduce chemical bonds to the aerogels to investigate the possible role of different interactions in our later work, as mentioned in the references (ref. 48, 49).

Comment 2: Also, what is the role of Ca⁺⁺? What are the comparison without Ca⁺⁺?

Response: The 3D printing of graphene oxide (GO) needs a gelator (or cross linker). In our case, Ca²⁺ ions play the role of gelator which makes GO dispersion into hydrogel. Without Ca²⁺, the size and shape of GO patterns cannot be fixed. We have added explanations about the role of Ca²⁺ and comparison without Ca ions gelator in supplementary information (Supplementary Figure 2 | Rheological behavior of printable GO-MWNT inks).

Comment 3: Some statements need to be rechecked and more accurate, one of the examples is “However, monolithic CAs are prone to lose structural integrity under deformations in either tensile or compression mode because of the extremely dilute connections and fragile joints in their porous network”, this is obviously not accurate since there are excellent results about the structural stability of graphene foam under even high compression (high elasticity), see Nature Comm, 2015, 6, 6141, Adv. Mater., 2016, 28, 3504, and related papers. I feel the refs about this part is over missed.

Response: Thanks for the comment. This sentence is just used to describe the situation before the reporting of compressible aerogels, and the following sentence “In the past decade, the compressive brittleness has been well resolved, affording various super-compressible CAs” has summarized the progress of compressible aerogels. We have revised the possible inaccurate statements (page 3, paragraph 1) and added related references in the revised manuscript (ref. 27, 28).

For Reviewer #3:

General comment: Improving the stretchability of carbon aerogels is one of the areas that are very important for several applications, namely in aerospace, smart robots, and wearable devices, among many others. Their use is however limited by the brittleness under tensile deformation. In addition, obtaining reversible extensive stretchability of such aerogels is also an unsolved problem.

This is extremely well written, interesting and thoughtful paper. Experiments are well conducted and the results are well documented. The conclusions are sound. The authors of the paper have clearly done a lot of work, and the results are entirely interesting.

The paper describes new way of making highly stretchable carbon aerogels using hierarchical synergistic assembly. These novel aerogels offer exceptional properties, very well described in the paper. The incorporation of 3D printing permits the design of various lattice structures, thus enabling different mechanical properties to be obtained. The described strategy of making these aerogels also enable detailed control of the hierarchical synergistic assembly. These results are important and the more so as they seem to involve different way of approaching the fabrication of carbon aerogels, unlike all other approaches (techniques) used and described before in the literature. The study has clearly very good practical significance. I have no hesitation in recommending this paper for publication.

Response: Thank you very much for your highly appreciated comment on our work.

REVIEWERS' COMMENTS:

Reviewer #1 made only remarks to the Editor and supports the publication of the manuscript.

Reviewer #2 (Remarks to the Author):

The revision is fine with me and my comments are essentially addressed well. I would suggest it to be accepted.

Responses to Reviewers

Reviewer #2:

The revision is fine with me and my comments are essentially addressed well. I would suggest it to be accepted.

Response: Thank you for your approval of our work.